# Development of Antibacterial Ti-Cu*_x_* Alloys for Dental Applications: Effects of Ageing for Alloys with Up to 10 wt% Cu

**DOI:** 10.3390/ma12234017

**Published:** 2019-12-03

**Authors:** Lee Fowler, Nomsombuluko Masia, Lesley A. Cornish, Lesley H. Chown, Håkan Engqvist, Susanne Norgren, Caroline Öhman-Mägi

**Affiliations:** 1Division of Applied Materials Science, Department of Engineering Sciences, The Ångström Laboratory, Uppsala University, Box 534, 751 21 Uppsala, Swedenhakan.engqvist@angstrom.uu.se (H.E.); susanne.norgren@angstrom.uu.se (S.N.); 2Advanced Materials Division, Mintek, Randburg 2124, South Africa; nomsombulukoh@mintek.co.za; 3School of Chemical and Metallurgical Engineering, and DST-NRF Centre of Excellence in Strong Materials, hosted by the University of the Witswatersrand, Johannesburg 2000, South Africa; lesley.cornish@wits.co.za (L.A.C.); lesley.chown@wits.co.za (L.H.C.)

**Keywords:** titanium alloys, copper, Ti_2_Cu, Ti_3_Cu, biomaterial, antibacterial

## Abstract

Peri-implantitis, a disease caused by bacteria, affects dental implants in patients. It is widely treated with antibiotics, however, with growing antibiotic resistance new strategies are required. Titanium-copper alloys are prospective antibacterial biomaterials, with the potential to be a remedy against peri-implantitis and antibiotic resistance. The aim of this study was to investigate Ti-Cu_x_ alloys, exploring how Cu content (up to 10 wt%) and ageing affect the material properties. Electron microscopy, X-ray diffraction, hardness testing, bacteriological culture, and electrochemical testing were employed to characterize the materials. It was found that alloys with above 3 wt% Cu had two phases and ageing increased the volume fraction of Ti_2_Cu. An un-aged alloy of 5 wt% Cu showed what could be Ti_3_Cu, in addition to the α-Ti phase. The hardness gradually increased with increased Cu additions, while ageing only affected the alloy with 10 wt% Cu (due to changes in microstructure). Ageing resulted in faster passivation of the alloys. After two hours the aged 10 wt% Cu alloy was the only material with an antibacterial effect, while after six hours, bacteria killing occurred in all alloys with above 5 wt% Cu. In conclusion, it was possible to tune the material and antibacterial properties of Ti-Cu_x_ alloys by changing the Cu concentration and ageing, which makes further optimization towards an antibacterial material promising.

## 1. Introduction

Peri-implantitis is a disease caused by bacterial population of an implant surface, which results in local inflammation of soft tissue and ultimately implant loosening and failure [1]. The loss of bone tissue is often associated with the inflammation surrounding the implants, leading to revision surgeries often being troublesome to successfully accomplish [1]. The prevalence of peri-implantitis in the general population is estimated to be between 19% [2] and 28% [3]. Despite the discrepancy in prevalence data, it is generally agreed that the prevalence is under-estimated at present [1,3]. Nevertheless, the importance of the problem is clear and increasing, considering that the dental implant industry is estimated to have reached USD 5 billion in 2018 [1]. The bacteria responsible for the disease include: *Peptostreptococcus micros, Porphyromonas ginigivalis and Treponema denticola*, among others [4,5]. The fact that these bacteria have virulent characteristic behavior [6], coupled with the increasing problems with antibiotic resistance [7], corroborate that an effective antibacterial solution is needed.

The effectiveness of bacterial colonization lies in the creation of biofilms [8], which have the ability to survive antibiotic treatments due to a multifaceted strategy of slower growth in the biofilm, poor penetration of antibiotics, and genetic variations in the bacteria [9,10]. The gram-positive *Staphylococcus epidermidis* is a biofilm forming bacteria, and is one of the most common bacteria in implant infections [11], making the bacteria a suitable strain for testing the antibacterial ability of biomaterials.

To date, much research has been focused on various approaches to reduce the bacterial burden correlated to implants, which include: surface treatments [12], the use of photochemical reactions [13], alloying of silver to titanium [14], and recently, the alloying of copper to titanium [15,16,17]. The antibacterial effect of Cu additions was found to be superior compared to silver [18]. The Ag appears to be effectual only at elevated temperatures [19], whereas Cu was found—using in vitro tests—to have antibacterial ability at ambient conditions [20]. For this reason, Cu is being investigated as an antibacterial element with increased frequency in thin films [21,22], ionic form [23,24], and in alloys [17,25,26]. In particular, Zhang et al. [27] have investigated the addition of Cu in titanium and found that there are discrepancies in the literature regarding required copper content and the mechanisms causing bacterial reduction. While success in antibacterial applications has been achieved with copper, the molecular process of bacterial death is not clear. However, it is hypothesized to be caused by Cu ions creating reactive oxygen species (ROS) via the Fenton reaction (${\mathrm{Cu}}^{+}\;+\;H_{2}O_{2}\;\to \;{\mathrm{Cu}}^{2+}\;+\;{\mathrm{OH}}^{-}\;+\cdot \mathrm{OH})$ [24], where ROS plays a role in the prohibition of bacteria 16SrRNA replication, resulting in bacteria death [28].

While it is important to understand the biological interactions in these systems, the antibacterial effect is only one aspect of successfully functioning antibacterial biomaterials. The mechanical properties are also important and should be considered, since implants experience mechanical loading in vivo. When excessive Cu is alloyed to Ti, intermetallic compounds (Ti_2_Cu [29] and Ti_3_Cu [30]) form, which can lead to brittleness [31]. While heat treatments of Ti alloys does allow some degree of ductility to be achieved in initially brittle materials, the resultant effect on microstructure and antibacterial properties must be assessed and understood. This inter-relationship between the various choices of compositions and heat treatments, and their impacts on mechanical properties and antibacterial effects, provide an opportunity for optimization of the final alloy. Therefore, the aim of the present study was to investigate these aspects (microstructural, mechanical and antibacterial) for different Ti-Cu alloys, to contribute to the understanding of these materials, and facilitate their future use as antibacterial biomaterials. The investigation only focused on alloys in the range 0 to 10 wt% Cu, since Cu ions released in this range of alloys did not show notable toxicity effects in previous works [15,23].

## 2. Materials and Methods 

### 2.1. Production and Preparation of Alloys

Alloys of titanium and copper were produced in a range from 0 to 10 wt% Cu (Table 1). Titanium grade 4 (Sandvik AB, Stockholm, Sweden), “CPTi”, and 99.9999% pure copper rods (365327-21.5G, Sigma Aldrich, St. Louis, MO, USA) were used to produce the alloy mixtures. Alloys were re-melted five times in an arc furnace, then re-melted and cast into rods in the same furnace (Series 5 Bell Jar, Centorr Vacuum industries, Nashua, New Hampshire, USA). Partial homogenization was achieved by turning over the melted alloys between each of the five melting events. Complete homogenization was achieved by heat treatments of the rods at 900 °C for 18 h, then 798 °C for 24 h and then finally, for half of the produced alloys, ageing at 400 °C for six hours. The annealing was done in vacuumed ampoules to reduce the oxygen contamination in the alloys, at a pressure of 1.333 mbar. All alloys were quenched into salt brine water by breaking the ampoules, to allow faster cooling. The alloy samples were cut into slices for analysis using a silicon carbide disk (150s, Struers, Ballerup, Denmark), to minimize cutting damage. Following sectioning, the samples were embedded in Bakelite resin (PolyFast, Stuers, Ballerup, Denmark) for metallographic preparation.

The metallographic preparation of the samples included grinding and polishing for subsequent microstructural studies. The three-step metallographic procedure developed by Vander Voort [32] was adapted for the present study, further details are given in Table 2.

### 2.2. Microstructural Studies

The microstructures of the materials were studied by scanning electron microscopy (SEM)—using a LEO1550 SEM (ZEISS group, Oberkochen, Germany) equipped with an INCA AZtec Energy Dispersive X-ray Spectroscopy (EDX) system (Oxford Instruments, High Wycombe, UK). Images were taken with an Everhart-Thornley (HE-SE2 detector, ZEISS group, Oberkochen, Germany) and backscattered electron detector (BSD). To ensure that the microstructure was discernible in the SEM, the samples were etched in Kroll’s Etchant: 2 mL HF, 4 mL HNO_3_ and 100 mL distilled water, with etching times of 30 s using cotton swabbing [33].

The energy dispersive x-ray spectroscopy (EDX) analysis of the 5Cu798 sample was performed on a Merlin SEM (ZEISS group, Oberkochen, Germany) equipped with an Ultim-Max 100 mm^2^ Silicon Drift Detector (Oxford Instruments, High Wycombe, UK) at a voltage of 15 kV and 8.5 mm working distance.

### 2.3. Hardness Studies

The hardness of the Ti-Cu*_x_* alloys was measured using a Vickers Hardness tester (Duravision EMCO, Prufmaschinen GmbH, Kuchl, Austria) with a 9.8 centinewton load. The machine was calibrated before testing with a standard Vickers sample, and an average was taken from three indentations.

### 2.4. Phase Identification

All samples studied by X-ray diffraction were polished as specified by Vander Voort [32] (Table 2). X-ray diffraction (XRD) was performed on a D8 Advance TWIN-TWIN diffractometer (Bruker, AXS GmbH, Karlsruhe, Germany) with Cu Kα radiation (Kα_1_ = 1.540598 Å with a Ni filter) and using the Bragg-Bretano experimental set-up. The detector system was a LynxEye XE PSD detector (Bruker, AXS GmbH, Karlsruhe, Germany). The EVA software suite (2015, Bruker) was used for phase analysis, while diffractogram plotting was done in Origin (2018b, OriginLab Corp., Northhampton, MA, USA). All crystallographic data were retrieved from the ICDD database PDF–4 + 2019 [34] and included PDF# 00-044-1294 (HCP-Ti), PDF# 00-015-0717 (Ti_2_Cu) [35] and PDF# 00-055-0296 (Ti_3_Cu).

### 2.5. Bacterial Luminescence by Direct Contact Test

The bacteria direct contact test has already been described [17], so only a summary is provided here. Staphylococcus epidermidis (XEN43) is a genetically modified bacterial strain that is bioluminescent due to the luxABCDE gene being bio-engineered into the bacterial genome [36,37]. Overnight bacterial inoculum of XEN43 was prepared a priori then seeded on the surface of sterile 5 mm diameter Ti-Cu*_x_* alloys (both aged and un-aged) and allowed to attach. Tryptic soy broth (TSB, Fluke-Sigma Aldrich, Stockholm, Sweden) was then carefully added to the test wells. Periodic measurements of luminescence were recorded for the samples on a Hidex plate CHAMELEON V (425-106, Multilabel counter, Turku, Finland), and the mean was used to determine the antibacterial rate (R) after two- and six-hours of exposure, using Equation 2 [38]:(1)$$R=\;\frac{N_{control}-N_{sample}}{N_{control}}\times \;100\%$$
where *N_control_* = mean luminescence from the CPTi sample, and *N_sample_* = mean luminescence for the Ti-Cu*_x_* alloys [38].

### 2.6. Corrosion Testing

The corrosion testing was performed on all the samples at 37 ± 1 °C in a phosphate buffered saline (PBS) solution (containing 8 g/L sodium chloride, 0.2 g/L potassium chloride, 1.44 g/L sodium hydrogen phosphate, and 0.24 g/L potassium di-hydrogen phosphate) maintained at a pH of 7.4.

The sample preparation for the corrosion studies included mounting the samples (CPTi and Ti–Cu_x_) of approximately 1 cm^2^ in area, in non-conductive epoxy resin with a copper wire soldered to the samples for electrical connection. The samples were ground to 120 grit surface finishes using a SiC paper, then rinsed with de-mineralized water and degreased in acetone.

The electro-chemical testing was performed using a computer-driven Potentiostat (AutoTafel, ACM Instruments, Cark, Cumbria, England). Two graphite rods acted as counter-electrodes and a Haber-Luggin capillary made the junction with a saturated calomel reference electrode (SCE). All potential values were with respect to the SCE. Nitrogen was bubbled continuously during testing in order to remove all the oxygen and maintain an anaerobic condition.

After each sample was immersed in the solution, the open circuit potential against time curve was recorded for up to 4 h to determine the open-circuit potential (OCP). When the potential had reached a sufficiently stable value at four hours, a cyclic polarization scan was recorded from −250 mV to 1500 mV versus the corrosion potential at a scanning speed of 10 mV/min. The scan direction was not reversed.

### 2.7. Statistical Analysis

The statistical analysis for the bacterial luminescence measurements was done in Origin software (2018b, OriginLab Corp., Northhampton, MA, USA) at two and six hours, using a One-way ANOVA with a Tukey HSD Post-Hoc test, and with Levene’s homogeneity of variance test. The same statistical test was also performed for the hardness with the exception that a Brown-Forsythe test for homogeneity of variance was performed. All tests had a statistical significance setting of *p* = 0.05.

## 3. Results

### 3.1. Microstructural Studies

Microstructures for the alloys were compared to determine any differences due to ageing. The microstructures of the alloys (both aged and un-aged) with less than 3 wt% Cu had a single-phase structure of α-Ti (the hcp solid solution of titanium) (Figure 1). However, the 1Cu798 showed peaks for the 2θ diffraction angle of the Ti_3_Cu phase, at 20.9° and 23.4° (Figure 2). Those with equal to and greater than 3 wt% Cu were all two-phased: Ti_2_Cu and α-Ti (Figure 1). X-ray diffraction studies confirmed these findings (Figure 2), where HCP-Ti and Ti_2_Cu peaks were identified, but peaks for the Ti_3_Cu crystals were also observed at the 2θ diffraction angles of 20.9° and 23.4° for the alloys: 1Cu798, 3Cu798, 5Cu798, and 3Cu400.

Comparison of the 10 wt% Cu before and after ageing displayed that a larger volume fraction of Ti_2_Cu precipitated due to this heat treatment. While an increase in Ti_2_Cu with ageing was also found for the 5 and 3 wt% Cu, the volume fraction of precipitated Ti_2_Cu was lower. In addition, these alloys had lamellar microstructures of Ti_2_Cu and α-Ti (Figure 1).

### 3.2. EDX Study of Precipitates

Precipitates in the 5Cu798 sample were studied to determine the presence of variations in the Cu content for the individual phases in the alloy. Since it is known that the Ti_2_Cu and Ti_3_Cu crystals have an atomic % Cu concentration of 33% and 25%, respectively [29,30], point analysis of individual crystals was performed to ascertain which phase was present. For sample area 1 (Figure 3a), the atomic concentrations for precipitates 1 and 2 were 33.2 ± 1.1 and 34.1 ± 0.3, respectively (Table 3). For sample area 2 (Figure 3b), the atomic concentrations for precipitates 3 and 4 were 31.5 ± 2.6 and 24.6 ± 0.5, respectively (Table 3).

### 3.3. Bacterial Luminescence

At two hours, no significance in luminescence was found among the samples, except between CPTi-798 and the 1Cu400 and 3Cu400 samples, which had significantly higher values (Figure 4). However, the antibacterial rate of 10Cu400 (R = 12%) was higher than the other aged alloys at this time point. All other alloys had a negative R at this time point, indicating that more bacteria were found on these alloys than on the respective control.

For the un-aged alloys, it was found that after six-hours of exposure (Figure 5) the R for 10Cu798 (42%) was higher than that for 5Cu798 (7%). The alloys with lower Cu content still had negative R-values. The 10Cu798 alloy also had significantly lower luminescence than alloys 1Cu798, 3Cu798 and 1Cu400 (*p* < 0.022). At this time point, the 10Cu400 had lower luminescence than the samples CPTi-798, 1Cu798, 1Cu400, 3Cu798, and 3Cu400 (*p* < 0.036). Within the aged alloys, it was found that R for 10Cu400 (45%) was greater than that for the 5Cu400 (15%). As for the un-aged alloys, the 3Cu400 and 1Cu400 had negative R-values.

### 3.4. Hardness Tests

The effects of alloying Cu to Ti and ageing on hardness were investigated (Figure 6). The 10Cu798 alloy had the highest hardness (350 ± 12 Hv) of the alloys while the CPTi-400 alloy had the lowest (120 ± 3 H_V_). The increased Cu addition to the CPTi correlated with a gradual augmentation in the Vicker’s hardness, except for the 10Cu798 alloy with a hardness twice that of the 5Cu798 alloy (171 ± 7 H_V_, *p* < 0.0001 in comparison with all other alloys). Ageing did not significantly affect the hardness of the alloys, except for 10Cu798, which was 1.9 times (*p* < 0.0001) harder than 10Cu400 (182 ± 1 H_V_), and CPTi798 which had a 14% (*p* < 0.021) increase in H_V_ after ageing.

### 3.5. Corrosion Tests

Corrosion studies were performed to determine the effect of the copper additions on the corrosion resistance, as well as the effect of ageing. The alloys were plotted in a single graph (Figure 7). The alloys aged at 400 °C showed a distinct corrosion profile after the anodic reaction took place. The aged alloys tended to passivate rapidly after the anodic reaction commenced. The un-aged alloys (quenched from 798 °C) clearly had a different corrosion profile trend, which indicated a gradual change in the passivation after the anodic reaction. 

## 4. Discussion

The present investigation focused on alloying up to 10 wt% of Cu into commercially pure titanium (a well-known implant material), where Cu has shown antibacterial potential in previous studies [17,20,38,39]. Its purpose was to determine the resultant microstructural and antibacterial characteristics of the materials, as a function of Cu addition and ageing heat treatments.

Microstructural and X-ray diffraction observations indicated no β-Ti in any of the alloys, despite a rapid quench (Figure 2). The reasons for this could be the active eutectoid transformation kinetics that drive transformation from $\beta \;-\;\mathrm{Ti}\;\to \;\alpha \;-\;\mathrm{Ti}\;+\;{\mathrm{Ti}}_{2}\mathrm{Cu}$ [40]. The two phases of α-Ti (HCP-Ti) and Ti_2_Cu were observed as predicted in phase calculations [17]. The CPTi and 1Cu–except the 1Cu798 alloy that could have Ti_3_Cu present−alloys were single-phase α-Ti alloys (Figure 1 and Figure 2), while in alloys with higher Cu content (aged and un-aged) both α-Ti and Ti_2_Cu were found. The ageing had a major effect on the microstructure and the Ti_2_Cu precipitates had a larger volume fraction in the aged alloys. The ageing of the 10Cu798 alloy coarsened the Ti_2_Cu from small precipitates to larger lamellar microstructures (Figure 1). A similar change occurred with ageing in the other two-phased alloys (5Cu798 and 3Cu798), but with coarsening of uniform lamellae of Ti_2_Cu in the α-Ti phase.

α-Ti and Ti_2_Cu were observed by X-ray diffraction, in addition to Ti_3_Cu at 2θ angles of 20.9° and 23.4° for the 1Cu798, 3Cu798, 3Cu400, and 5Cu798 alloys. EDX measurements on a 5Cu798 sample confirmed that some of the precipitates had the expected 33 at% Cu for Ti_2_Cu [35], while at least one precipitate had compositions closer to 25 at% Cu, which is the expected composition for Ti_3_Cu [30]. However, the results might have been affected by analyzing small areas, which resulted in collecting the signal from the surrounding phases. Although this study did not make use of a standard reference material for the analysis, the results gave insight into the variation of composition for the precipitates with Cu in the Ti-Cu*_x_* alloys. It should however be noted that the precipitates with compositions closer to Ti_3_Cu were in lower volume fractions—for the 5Cu798 alloy—relative to the precipitates with compositions close to 33 at% Cu. This lower volume fraction could be the reason for the absence of Ti_3_Cu by X-ray diffraction in the 10 wt% Cu alloys, since it could have been below the detection limit for the technique. However, this was only a preliminary study on one alloy and further studies are needed to confirm the general presence of Ti_3_Cu in Ti-Cu*_x_* alloys.

As well as the microstructure, the hardness was also affected by Cu additions and ageing, and the 10Cu798 was twice as hard as the 10Cu400 alloy. The mechanism causing this could be the coarser lamellar structure in the 10Cu400 [41]. Ageing had a hardening effect on the CPTi-798 alloy as well, because the hardness was significantly higher for the CPTi-400 alloy.

The bacteria test gave an interesting result after two hours of exposure for the alloys, and 10Cu400 had an antibacterial rate R of 11%, while all the other Ti-Cu*_x_* alloys had higher mean luminescence than CPTi. The reason for the higher luminescence could be that the bacteria underwent a stress response, which caused a rapid increase in population [23]. Comparison of the aged alloys at two hours and at six hours gave a similar trend for antibacterial rate (R). The 10Cu400 alloys seemed to be effectual at bacteria reduction already from two hours and thereafter increased, and at six hours the R was 46%. The 5Cu798 and 5Cu400 alloys after six hours had R values of 7% and 15%, respectively, although this was not significantly different from alloys with lower Cu content. The greater R at six hours for both aged (45%) and un-aged (42%) 10 wt% Cu alloys agreed with previous findings where a higher Cu content has been reported to yield a stronger antibacterial material [17,38]. The difference in R for the 10 wt% Cu alloys at two hours could be due to the larger amounts of Ti_2_Cu in the aged alloy leading to higher Cu ion release rates as has been observed in studies on Ti-6Al-4V-5Cu alloys [28]. However, further studies on ion release from Ti-Cu*_x_* are needed to support this conclusion. Additionally, direct contact of bacteria with a Cu rich surface—such as the 10Cu400 with higher volume fraction Ti_2_Cu—could also have played a role in the antibacterial effect [24]. Thus, Cu ions and direct contact killing contributed to the antibacterial effect, and future studies could investigate how the amount and size of the Ti_2_Cu phase influences copper ion release and the contact killing of bacteria.

The corrosion results indicated that alloying with Cu and ageing affected the corrosion rates and passivation rates. Alloying and ageing induced faster passivation for higher Cu contents, and decreased the corrosion rates. These results are in agreement with the findings of Zhang et al. [38] for an aged 4 wt% Cu alloy.

A limitation to the present study is that only alloys in the range of 0 to 10 wt% Cu were investigated. It has been demonstrated that a higher Cu content yields a better antibacterial response [17], however, an excess of copper can be harmful to the human body [42]. Previous studies [15,23] have reported that the Cu ions released from alloys with up to 10 wt% Cu did not have any notable toxic effects and were therefore the focus of the present study. Moreover, only two heat treatments were studied due to a necessary delimitation of the study. There are other heat treatments that could be tested, and future studies should further optimize the alloying process. The investigation on the presence of Ti_3_Cu was a preliminary study and the results should not be viewed as conclusive due to only one alloy being characterized, the low number of precipitates characterized and the low resolution of the EDX technique. Another limitation is that the antibacterial ability of the alloys was only tested with one type of bacteria. However, the bacterium used, Staphylococcus epidermidis*,* is one of the most prevalent bacteria in implant infections and its use additionally allowed comparison with previous studies [23].

## 5. Conclusions

Both Ti_2_Cu and Ti_3_Cu were detected in Cu alloys using X-ray diffraction. A preliminary EDX study on the un-aged 5 wt% Cu alloy also indicated the presence of Ti_3_Cu, however with the Ti_2_Cu in majority. While the study of the Ti_3_Cu was not quantitative in this investigation, it is recommended that future works focus on determination of this phase.

The hardness was reduced after ageing of the 10Cu798 alloy, as the microstructure changed from small precipitates to coarser lamellar microstructures. The alloys had good antibacterial rates above 5 wt% Cu (aged and un-aged) after six-hours of exposure, and the two-hour exposure was probably too short a duration for bacterial reduction to occur in most of the alloys. Ageing the alloys only ensured faster antibacterial rates after 2 hours, when the concentration of Cu was greater than 10 wt% Cu. The addition of Cu to Ti and ageing increased the corrosion resistance of the alloys, which could protect the biomaterial in vivo. However, since the ageing also lowered the hardness of the higher Cu content alloys, care should be taken to avoid over-ageing, and hence softening.

In conclusion, the un-aged 10 wt% Cu alloy was considered a suitable candidate material, which provided a good antibacterial effect, with superior hardness and corrosion protection. The ideal composition and heat treatment of these materials will however depend on the specific application envisioned, and will require further optimization.

## Figures and Tables

**Figure 1 materials-12-04017-f001:**
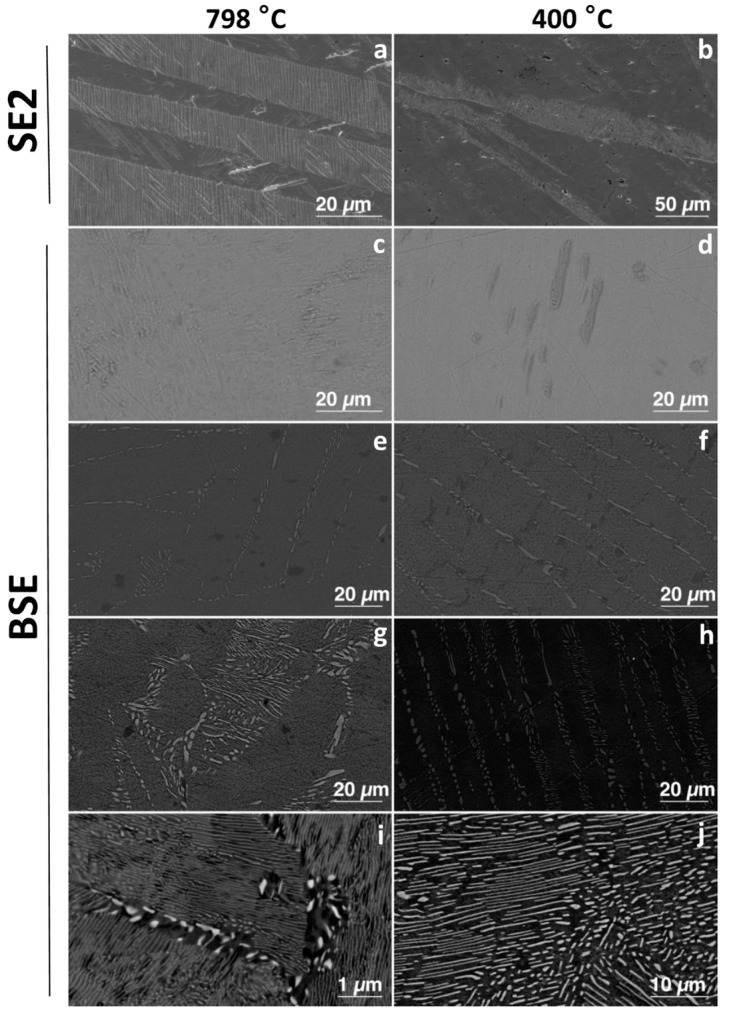
Secondary electron images of: (**a**) CPTi-798, (**b**) CPTi-400, and backscattered electron images of: (**c**) 1Cu798, (**d**) 1Cu400, (**e**) 3Cu798, (**f**) 3Cu400, (**g**) 5Cu798, (**h**) 5Cu400, (**i**) 10Cu798, and (**j**) 10Cu400.

**Figure 2 materials-12-04017-f002:**
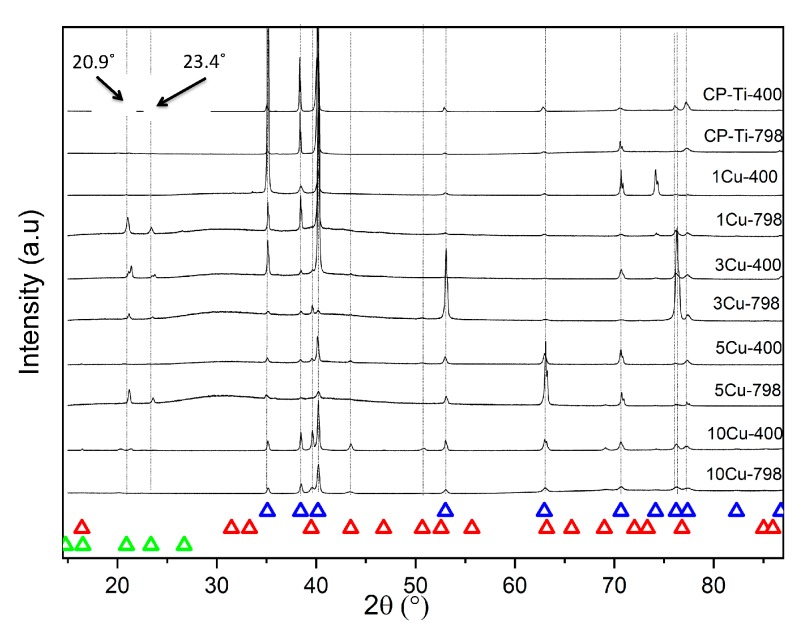
X-ray diffraction patterns of Ti-Cu*_x_* alloys heat-treated at 798 °C, some aged at 400 °C: blue triangles indicate HCP-Ti, red triangles indicate Ti_2_Cu and green triangles indicate Ti_3_Cu. Note the peaks for Ti_3_Cu at 20.9° and 23.4°.

**Figure 3 materials-12-04017-f003:**
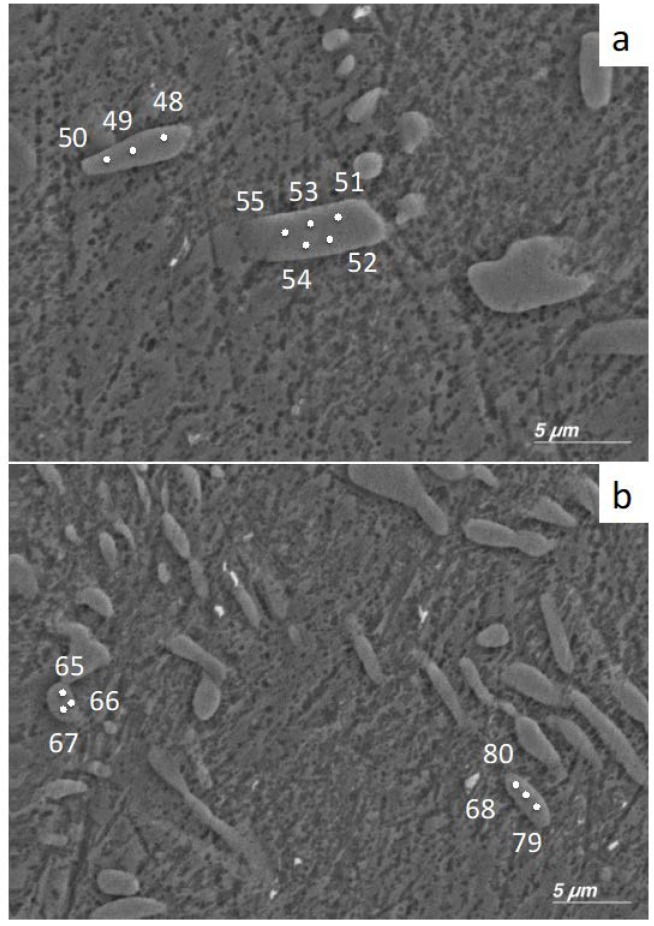
Positions of EDX spectral analyses of sample 5Cu798: (**a**) Sample area 1 with precipitate 1 (48, 49, 50) and precipitate 2 (51, 52, 53, 54, 55). (**b**) Sample area 2 with precipitate 3 (65, 66, 67) and precipitate 4 (68, 79, 80).

**Figure 4 materials-12-04017-f004:**
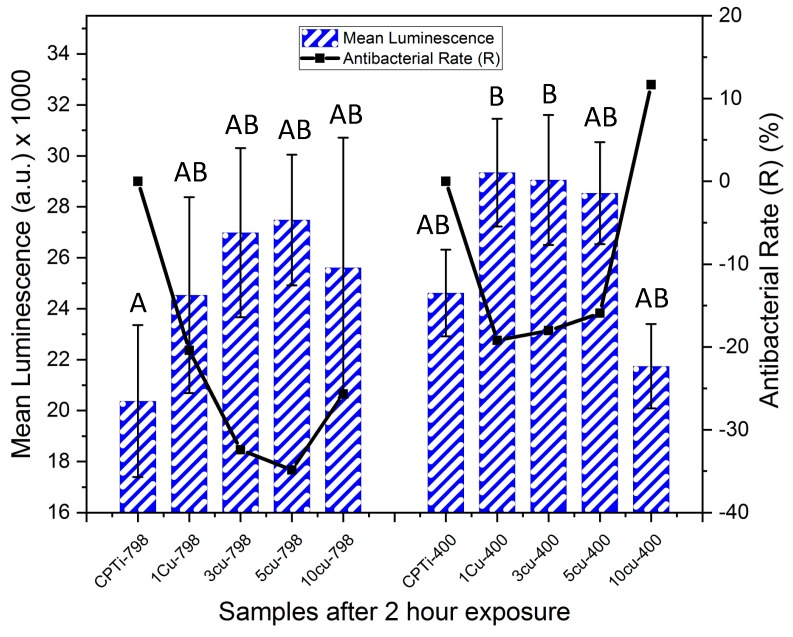
Mean luminescence counts (bars for standard deviation) of XEN 43 bacteria after two hours for all alloys, the negative controls were (CPTi), and the corresponding antibacterial rates are shown in black. Levene’s Test: F (9,20) = 1.83, *p* = 0.123. One-way Anova: F = 3.176 (*p* = 0.015) with Tukey Test. Note: The same letters (e.g, “A” compared to “A”) denotes being non-significantly different, while different letters (e.g. “A” compared to “B”) denotes being statistically significantly different.

**Figure 5 materials-12-04017-f005:**
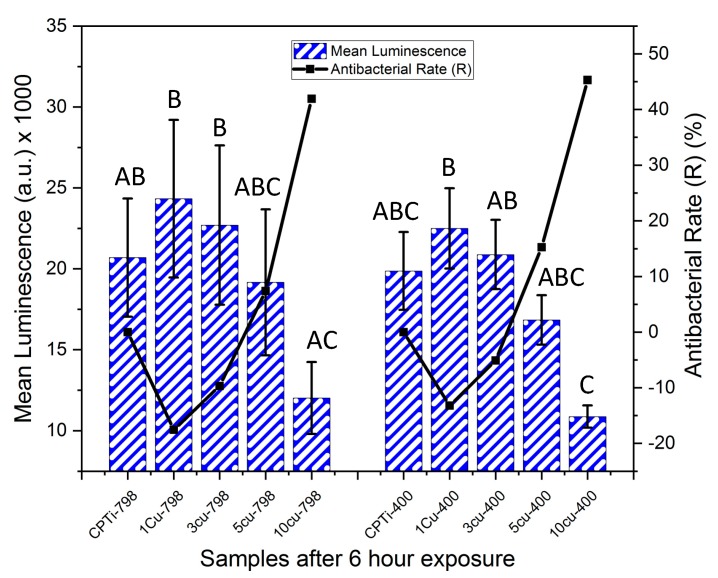
Mean luminescence counts (bars for standard deviation) of XEN 43 bacteria after six hours for all alloys, the negatives controls were CPTi, and the corresponding antibacterial rates are shown in black. Levene’s Test: F (9,20) = 2.21, *p* = 0.066. One-way Anova: F = 5.697 (*p* < 0.001) with Tukey Test. Note: The same letters (e.g, “A” compared to “A”) denotes being non-significantly different, while different letters (e.g. “A” compared to “B”) denotes being statistically significantly different.

**Figure 6 materials-12-04017-f006:**
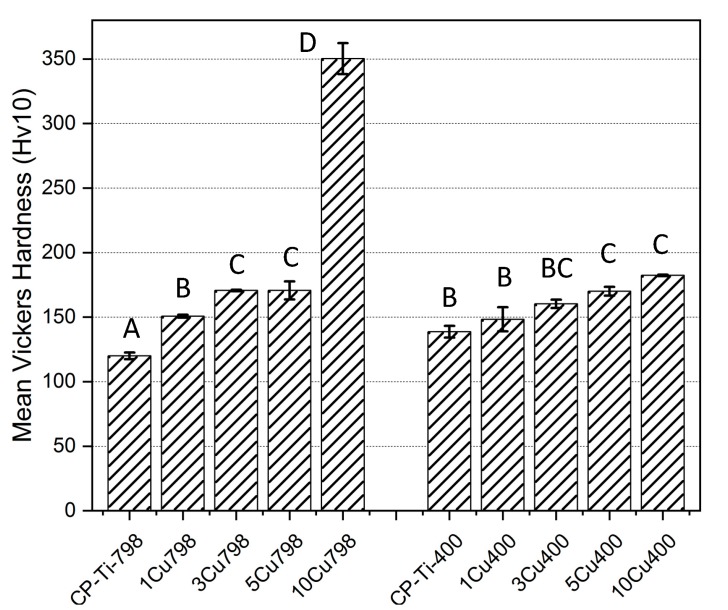
Mean Vickers hardness (bars for standard deviation) of all samples. Brown-Forsythe: F (9,20) = 1.413, *p* = 0.2476. One-way Anova: F = 368.92 (*p* < 0.001) with Tukey test. Note: The same letters (e.g, “A” compared to “A”) denotes being non-significantly different, while different letters (e.g. “A” compared to “B”) denotes being statistically significantly different.

**Figure 7 materials-12-04017-f007:**
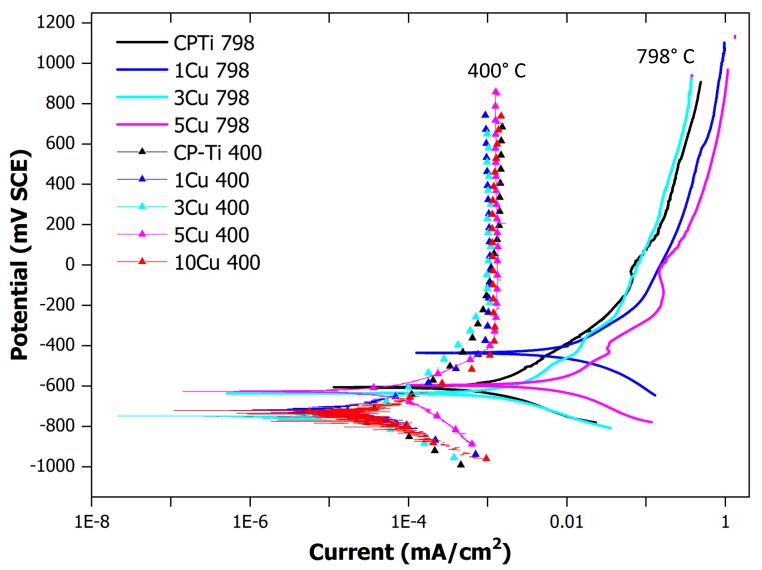
Corrosion plots for the aged (at 400 °C) and un-aged (quenched from 798 °C) alloys. The aged alloys are indicated with triangles to indicate the profile. Note: the 10Cu798 alloy was not included in the corrosion study.

**Table 1 materials-12-04017-t001:** Ti-Cu*_x_* samples used in the study.

Nominal wt% Cu	CPTi	1 wt% Cu	3 wt% Cu	5 wt% Cu	10 wt% Cu
Alloys heat treated at 798 °C (un-aged)	CPTi-798	1Cu798	3Cu798	5Cu798	10Cu798
Alloys aged at 400 °C	CPTi-400	1Cu400	3Cu400	5Cu400	10Cu400

**Table 2 materials-12-04017-t002:** Three-step metallographic preparation of Ti-Cu (All products sourced from Struers, except H2O2 sourced from BASF SE, Ludwigshafen, Germany).

Steps	1–Grind	2–Rough Polish	3–Final Polish
Surface	SiC–320P	MD–Dur cloth	MD–Floc cloth
Abrasive	-	6 µm diamond suspension	OP-S Si-Colloids and H_2_O_2_ (5:1) solution
Lubricant	Water	DP Lubricant Red	-
Speed (rpm)	200 contra	150 contra	150 contra
Duration (min)	Until planar	15 min	15 min

**Table 3 materials-12-04017-t003:** EDX point spectral analyses of sample areas 1 and 2 (point spectral analysis correspond to points in Figure 3).

Precipitate (EDX Point Analyzed)	Atomic % Copper (Balance % Titanium)	Possible Phases
Precipitate 1 (48, 49, 50)	33.2 ± 1.1	Ti_2_Cu
Precipitate 2 (51, 52, 53, 54, 55)	34.1 ± 0.3	Ti_2_Cu
Precipitate 3 (65, 66, 67)	31.5 ± 2.6	Ti_2_Cu
Precipitate 4 (68, 79, 80)	24.6 ± 0.5	Ti_3_Cu

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
