# Peer review of "Development of Antibacterial Ti-Cux Alloys for Dental Applications: Effects of Ageing for Alloys with Up to 10 wt% Cu"

_materials, 2019, doi:10.3390/ma12234017_

Round 1
Reviewer 1 Report
This article has the aim to investigate the aspects for different Ti-Cux alloys as antibacterial biomaterials and to contribute the deeper understanding of these materials, and facilitate their future use.
The actuality of the work is determined by the choice of the research object (alloys of the Ti-Cu system) as a promising biomedical material with an improved set of operational properties.
The following comments should be made as general comments on the text of the file.
The text is overloaded with redundant information, often of purely encyclopedic content.
- For example, microhardness measurement is a standard procedure and in my opinion does not require a description in a separate
- The Discussion section in its first part is not analytic in nature but practically repeats the division 3 Results paragraph.
- The data presented in Figure 3 (analysis of the zones of individual precipitates) and the subsequent conclusions on the phase composition of these precipitates are not entirely obvious since the analysis zones, as a rule, are several microns, which is comparable to the size of the precipitates. And this means that the matrix material (based on α-Ti) may also be included in the analysis.
- In the Conclusion section, recommendations on the optimal alloy composition and heat treatment regimes are not made. If the title says “Development of ………alloys…..” , then the conclusion should be given recommendations.
Author Response
The authors would like to thank the Reviewer for the valuable comments and feedback on our submitted manuscript, which we feel, and hope, have improved the quality of the work. The Reviewer’s comments are answered point-by-point below (Reviewer comments are marked in black, and our responses in red) and all the resulting changes are highlighted in the submitted manuscript in yellow.
Comments and notes by reviewer 1:
This article has the aim to investigate the aspects for different Ti-Cux alloys as antibacterial biomaterials and to contribute the deeper understanding of these materials, and facilitate their future use.
The actuality of the work is determined by the choice of the research object (alloys of the Ti-Cu system) as a promising biomedical material with an improved set of operational properties.
The following comments should be made as general comments on the text of the file.
The text is overloaded with redundant information, often of purely encyclopedic content.
Comments:
For example, microhardness measurement is a standard procedure and in my opinion does not require a description in a separate
Response: Thank you for the comment. The hardness explanation has been adapted to exclude redundant information.
Comments:
The Discussion section in its first part is not analytic in nature but practically repeats the division 3 Results paragraph.
Response: Thank you for the comment. The first section of the Discussion has been changed to the following (Line: 256-259):
The initial text:
“Alloying copper in commercially pure titanium (a well-known implant material) has potential in the pursuit of developing antibacterial materials, due to the effective antibacterial ability of copper [20]. However, the effect of copper on the microstructures and the material properties needs to be understood, so that the alloys can be optimized. “
Has been replaced with:
“The present investigation focused on alloying Cu into commercially pure titanium (a well-known implant material), where Cu has shown antibacterial potential in previous studies [17,20,38,39]. The purpose was to determine the resultant microstructural and antibacterial characteristics of the materials, as a function of Cu addition and ageing heat treatments.”
Comments:
The data presented in Figure 3 (analysis of the zones of individual precipitates) and the subsequent conclusions on the phase composition of these precipitates are not entirely obvious since the analysis zones, as a rule, are several microns, which is comparable to the size of the precipitates. And this means that the matrix material (based on α-Ti) may also be included in the analysis.
Response: Thank you for the comment. It is true that the matrix effects play a role in the EDX analysis. It is also true that the interaction volume for the beam is 1 μm3, which limits the resolution for differentiating between various chemical phases. For this reason, the present analysis specifically focuses on qualitative, and not quantitative information. Therefore, the emphasis is on changes in composition, not determining the exact composition of each precipitate. Furthermore, to avoid experimental errors, a statistical average is obtained by 3 point measurements on random locations approximately 1 μm apart.
A superior technique for quantitative purposes is to use wavelength dispersive x-ray spectroscopy (WDX), with a standard for materials containing Cu. This is the aim of future work on these materials, and the present analysis should be regarded as a first attempt to study the Ti3Cu phase. Furthermore, a sentence was added to the discussion highlighting that the results are only preliminary and should be viewed with caution.
Comments:
- In the Conclusion section, recommendations on the optimal alloy composition and heat treatment regimes are not made. If the title says “Development of ………alloys…..” , then the conclusion should be given recommendations.
Response: Thank you for the comment. The conclusion has been adapted to include the information suggested. The following text has been added to the conclusion section (Line 325-342):
“Both Ti2Cu and Ti3Cu were detected in the alloys using X-ray diffraction. A preliminary EDX study on the un-aged 5wt%Cu alloy also indicated the presence of Ti3Cu, however with the Ti2Cu in majority. While the study of the Ti3Cu was not quantitative in this investigation, it is recommended that future works focus on determination of this phase.
The hardness was reduced after ageing of the 10Cu798 alloy, as the microstructure changed from small precipitates to coarser lamellar microstructures. The alloys had good antibacterial rates above 5 wt%Cu (aged and un-aged) after six-hours of exposure, and the two-hour exposure was probably too short a duration for bacterial reduction to occur in most of the alloys. Ageing the alloys only ensured faster antibacterial rates after 2 hours, when the concentration of Cu was greater than 10 wt%Cu. The addition of Cu to Ti and ageing increased the corrosion resistance of the alloys, which could protect the biomaterial in vivo. However, since the ageing also lowered the hardness of the higher Cu content alloys, care should be taken to avoid over-ageing, and hence softening.
In conclusion, the un-aged 10 wt%Cu alloy was considered a suitable candidate material, which gave good antibacterial effect, with superior hardness and corrosion protection. The ideal composition and heat treatment of these materials will however depend on the specific application envisioned, and will require further optimization.”

Reviewer 2 Report
1) Figure 1 should present the comparative SEM images using the same scale in order to have a better overview (see a, b and i, j).
2) EDX study should comprise more samples, the existent one is limited to only one sample (5Cu798).
Author Response
The authors would like to thank the Reviewer for the valuable comments and feedback on our submitted manuscript, which we feel, and hope, have improved the quality of the work. The Reviewer’s comments are answered point-by-point below (Reviewer comments are marked in black, and our responses in red) and all the resulting changes are highlighted in the submitted manuscript in yellow.
Comments and suggestions from Reviewer 2:
Comment:
Figure 1 should present the comparative SEM images using the same scale in order to have a better overview (see a, b and i, j).
Response: Thank you for the comment. The overall purpose of Figure 1 was to display the changes of the microstructures of the alloys due to increases in wt%Cu and the ageing heat treatments at 400Ë™ C.
Since the ageing heat treatments generally cause crystal growth in the materials, displaying the microstructures before and after ageing at the same magnification does not achieve the desired result of showing the microstructural changes for the Ti-Cu alloys. For this reason the magnifications suggested by the reviewer will not allow the desired information to be displayed in Figure 1, and we suggest to keep the figure as it is. We hope that this will be acceptable to the reviewer.
Comment:
2) EDX study should comprise more samples, the existent one is limited to only one sample (5Cu798).
Response: Thank you for the comment. The use of EDX to study the Ti3Cu phase in the 5Cu798 alloy was performed to qualitatively determine the possible presence of this phase in the material, and to compare this result to other studies where Ti3Cu was not reported.
In the present study the use of several point analyses was used to get an average of the chemical composition (which was qualitative since no standard samples was used for comparison). For this reason performing the same analysis on various other alloys in this study would not increase the qualitative information gained.
We however recommend in the manuscript that future work be focused on gaining quantitative information of the Ti3Cu phase.

Reviewer 3 Report
The goal must be more specific more objective. The author refer “aspects and understand the materials”: what aspects, understand how?
How was the sample calculation performed?
p statistics are italic
What are the question marks in figures 4 and 5 and 6?
The authors should include a paragraph with study limitations
The authors should talk about the effects of Ti and Cu excess on the body. Will they be released from these leagues? if so, can they be used without risk to patients?
Conclusions should be more objective. Part of the conclusions are in fact discussion.
Author Response
The authors would like to thank the Reviewer for the valuable comments and feedback on our submitted manuscript, which we feel, and hope, have improved the quality of the work. The Reviewer’s comments are answered point-by-point below (Reviewer comments are marked in black, and our responses in red) and all the resulting changes are highlighted in the submitted manuscript in yellow.
Comments and suggestions from Reviewer 3:
The goal must be more specific more objective. The author refer “aspects and understand the materials”: what aspects, understand how?
Response: Thank you for the comment. Additional information has been added to the indicate the aims of the study (Line 79-81):
The initial text:
“Therefore, the aim of the present study was to investigate these aspects for different Ti-Cu alloys, to contribute to the understanding of these materials, and facilitate their future use as antibacterial biomaterials.”
Has been replaced with:
“Therefore, the aim of the present study was to investigate these aspects (microstructural, mechanical and antibacterial) for different Ti-Cu alloys, to contribute to the understanding of these materials, and facilitate their future use as antibacterial biomaterials.”
Comment:
How was the sample calculation performed?
Response: Thank you for the comment. Assuming the reviewer is referring to the compositions in Table 1: The exact compositions for the materials were not determined but instead, the nominal compositions of the alloys are provided in the table. These compositions were determined by weighing the individual masses of the commercially pure titanium and the added copper prior to alloying in an arc furnace. Between each melting step, the samples were weighed to ensure the sample mass remained the same throughout.
Comment:
p statistics are italic
Response: Thank you for the comment. The “p” in the statistical reporting has been changed to the italic “p” as recommended.
Comment:
What are the question marks in figures 4 and 5 and 6?
Response: There are no question marks in these figures so there must be a problem with compatibility between word processors. If the reviewer is referring to the letters “A”, “B”,etc: These letters are used to indicate samples that are non-significantly different and those that are significantly different, based on the ANOVA test with Tukey testing applied. The following is from the text in the manuscript:
“Note: The same letters (e.g, “A” compared to “A”) denotes non-significantly different, while different letters (e.g. “A” compared to “B”) denotes statistically significantly different.”
Comment:
The authors should include a paragraph with study limitations
Response: Thank you for the comment. A section on limitations have been added to the end of the Discussion section.
Comment:
The authors should talk about the effects of Ti and Cu excess on the body. Will they be released from these leagues? if so, can they be used without risk to patients?
Response: Thank you for the comment. The risk of toxicity due to Cu ions has been addressed in the Introduction section with the following statement (Line 81-83):
“The investigation focused on alloys in the range 0 to 10 wt%Cu only, since Cu ions released in this range of alloys did not show notable toxicity effects in previous works [15,23].”
Comment:
Conclusions should be more objective. Part of the conclusions are in fact discussion
Response: Thank you for the comment. The conclusions have been made more objective and concise to remove the discussion sections from the conclusion. Furthermore, a recommendation on the optimal alloy composition has been added to the Conclusion section.
